# Energy-Efficient Inference with Small Language Models: A Comparative Study on Code Generation, Classification, and Environmental Impact

## Abstract

Large language models (LLMs) are widespread in enterprise applications for code completion, email classification, and sentiment analysis. Although these models have good performance, their high computational requirements make them consume high energy in inference. Can smaller language models (SLMs) with three billion parameters (Qwen2.5-3B-Instruct) perform similarly in structured high-frequency tasks while providing significantly lower environmental impact?

We tested an SLM on three enterprise workloads: code generation with HumanEval benchmark (164 tasks), HR email routing (1,339 examples), and binary sentiment analysis (872 samples from SST-2). We recorded output quality, inference latency, throughput, GPU memory utilisation and energy consumption using direct power measurements via NVIDIA SMI. For controlled comparison, we also evaluated a 14-billion-parameter LLM (Phi-3-medium-4k-instruct, 4-bit quantized) under identical hardware and experimental conditions on the same three tasks. The SLM achieved 73.2% Pass@1 on code generation, 90.5% on sentiment analysis, and 79.9% on email classification. Under identical conditions, the SLM consumed 3–11× less energy per query than the 14B LLM across all tasks, while achieving higher accuracy. When compared against cloud-deployed frontier models using published inference cost estimates (Jegham et al., 2025), the energy reduction scales to 388–1,333×, reflecting compounded effects of model scale and datacenter overhead. Scaling to organisational context, replacing a locally-hosted 14B LLM with an SLM for 10,000 daily code completions, 100,000 sentiment queries, and 50,000 monthly email classifications would save approximately 5,500 kWh annually, reducing $CO_2$ emissions by 2.4 metric tons. Savings relative to cloud-deployed frontier models would be proportionally larger. An SLM-first deployment strategy is a practical way to attain sustainable AI with significant energy savings.

## 1 Introduction

Huge language models also penetrated the software development process and enterprise automation with remarkable speed. Code completion with aid of AI is used by developers, whereas such models find application in organisations to route emails, carry out sentiment analysis, and classify documents. There are common features of these applications; they are continuous, take large amounts of incoming requests, and generally produce succinct but compact output.

Recent works have done little beyond investigating the energy cost of model training and concluded that large model requires considerable computational resources to train. However, the inference energy usage is the biggest part of the total lifecycle footprint after the deployment of the models (Jegham et al., 2025). The inference operations are linear with any number of execution and query, compared with training in which inference is performed only after one time. Per-request inefficiencies that are minor at any rate can scale to huge environmental and operational expenditures since a single organisation can handle the inference requests (often millions per day).

This scenario brings a basic mismatch: general-purpose models are used in most high-frequency enterprise work, despite any simple and deterministic required outputs. Activities like those of deriving functions based on specifications, sorting emails in given categories or even the binary sentiment classification take up a narrow output space and are very predictable. The characteristics of these models imply that task-specific models with orders of magnitude fewer parameters can support similar performance levels while achieving significant reduction in computational cost.

The primary question the present work is built on is whether it is possible to achieve parity in the quality of small language models in the context of structured and repeatable tasks and achieve a considerable cutback in the amount of energy that is used during the inference. This would confirm an SLM-first implementation paradigm: small models would be deployed by default and only when more problem complexity or ambiguity require more capacity should bigger models be increased.

## 2 Problem Statement

The modern-day use of AI seems ineffective: organisations utilize large-scale general-purpose language models to solve the problems that do not justify the entire capacity of the model or the ongoing computational cost. In this paper, a study on the use of small language models (around three billion parameters) to facilitate common enterprise workloads and produce orders of magnitude improvements on inference energy costs is explored.

There are three representative types of tasks whose evaluation is performed:

**Code Generation:** The developers would demand an implementation of the functions depending on the explicit demands and test them against the predefined unit tests. Such jobs require syntactic and logical accuracy under a clearly defined group of constraints.

**Email Classification:** The enterprise system classifies incoming traffic to a set type by either department or priority level. Space used for labeling is fixed and predictions must be singleton.

**Sentiment Analysis:** The sentiment of customer review, products and surveys are categorized into binary or ternary customer sentiment. The encoding into discrete groups produced by such activities does not mean that text is generated in a simple manner.

Each of the three types has traits that are similar to the abilities of smaller models: structured inputs, constrained outputs, low semantic ambiguity and high query frequency. A transition into smaller models can turn out to be quite practical and even useful when it comes to the environmental and economic impact of performing such tasks on the organisational level.

## 3 Contributions

This manuscript contributes to the existing body of literature on the topic of energy efficient deployment of language models in the following ways:

1. **Empirical Study of SLM Performance:** We verify and lower bound performance of a 3-billion parameter language model execution on a set of three overall enterprise tasks and we measure both quality indicators (Pass@1 for code and classification accuracy) and resource indicators (latency, throughput, memory usage).

2. **Controlled LLM Baseline:** We evaluate a 14-billion-parameter LLM (Phi-3-medium-4k-instruct, 4-bit quantized) under identical hardware, dataset, and decoding conditions on all three tasks, enabling direct and fair comparison between the SLM and a substantially larger model.

3. **Direct Energy Measurement:** We replace proxy-based energy estimation with real-time power measurements via NVIDIA SMI (100 ms polling interval), reporting actual watts, joules, and watt-hours per query for both the SLM and LLM baselines.

4. **Task Suitability Framework:** We identify task properties that define the feasibility of the SLM e.g., output organization, length constraints and entropy and offer deployment decision guidelines.

5. **Environmental Impact Estimation:** We estimate the tangible energy savings, and CO2 reductions which can be achieved under SLM-first deployment strategies and prove the instant sustainability rates with the absence of any infrastructural modifications.

6. **Hybrid Deployment Strategy:** We suggest an escalation procedure, where the SLM is involved in managing standard queries and shifting ambiguous or difficult queries to bigger models, which is optimal in terms of the accuracy versus efficiency trade-off.

7. **Reproducibility:** We have supportive replication, indicating that we give fully trained empirical models, available in Google Colab notebooks that can be replicated and altered to fit other situations.

# 4 Related Work

## 4.1 Code Generation Benchmarks

HumanEval benchmark proposed by Chen et al. (2021) developed a standardized structure of testing python programming synthesis skills against 164 problems along with sources of unit tests. Pass@$k$ measure was used which is the likelihood of at least one program sampled passing tests. Later benchmarks such as MBPP (Austin et al., 2021) and APPS (Hendrycks et al., 2021) broadened both the scale diversity and language diversity, and difficulty hypotheses. Most of the studies that have been conducted here focused heavily on functional correctness without paying much attention to computation efficiency. Recent publications (Ashraf et al., 2025) have bridged this gap by quantifying energy consumption as well as accuracy by proving what smaller models that can make correct outputs always use less energy than larger ones, hence achieving improvements in efficiency without compromising quality.

## 4.2 Small Language Models

While no universally accepted boundary separates small from large language models, we adopt the practical threshold of 10 billion parameters, consistent with recent surveys of the SLM landscape (Soni et al., 2025). Small language models that are traditionally defined as having fewer than ten billion parameters have become effective replacements of large general-purpose models in task-specific contexts. Models like Qwen2.5-Coder, CodeLlama variants, and Phi family are also able to deliver a competitive performance on the targeted tasks but do so consuming significantly fewer resources in terms of computational resources (ArXCompass, 2024). Importantly, both the 3B SLM and the 14B LLM evaluated in this work fit within the 16 GB VRAM of a single NVIDIA T4 GPU under 4-bit quantization, enabling a controlled comparison on identical hardware. Research on SLMs concentrates more on domain specific applications, demonstrating that with proper fine-tuning or prompt engineering, such models can be able to be equivalent and at times superior to larger systems in specialized tasks. However, the systematic cross-task comparisons especially on the use of energy among various workloads are under-researched.

## 4.3 Enterprise Automation through NLP

The functions implemented in enterprise NLP include email filtering, spam detection, and sentiment analysis. The classical machine-learning methods, such as gradient-boosted trees and Naïve Bayes, have been compared to transformer-based systems and sentence transformers. Big models have performed well in few-shot, as well as, low-resource conditions (Labonne & Moran, 2023). In particular comparisons of GPT-4 with domain-specific models including BERT and XLNet document that smaller model types can equal or out-perform LLMs on less general data, like social media text, and in considerably cheaper circumstances (Zhang et al., 2023). The above observations highlight the high impact of model adaptation on predictive performance and efficiency.

### 4.4 Sustainability and Energy of AI System

AI systems have also made the environmental footprint and energy footprint critical concerns when it comes to large-scale deployment. Studies are being done to quantify the energy, water and carbon footprint of inference of LLM inference in different configurations. It has been shown that the output of a single query to a commercial LLM can use many watt-hours, and combined usage can be disastrous at scale (Jegham et al., 2025). Although typically most studies on sustainability are training costs, inference energy, in many cases, will be the most prominent lifecycle consumption of deployed systems (Fernandez et al., 2025). However, the existing literature on systematic task-level comparisons of energy use when optimising a deployed system to be sustainable is limited, and this is a major gap in knowledge. The current paper has outlined this shortcoming by presenting task-specific energy comparative curves of small and large models and organizational scale of impact projections, thus offering evidence-based decision supports in selection of models in production settings where sustainability and operational costs are the major factor.

## 5 Methodology

### 5.1 Model Selection

The primary SLM evaluated is Qwen2.5-3B-Instruct, selected for its instruction-oriented nature which is friendly to prompt-following tasks and its balance of capacity and efficiency. This model has approximately three billion parameters, falling within the SLM regime for exploring performance on routine enterprise tasks.

As a controlled LLM baseline, we evaluate Phi-3-medium-4k-instruct (14B parameters), also loaded under 4-bit quantization on the same NVIDIA T4 GPU. Both models are evaluated under identical conditions: same datasets, same random seeds, same greedy decoding configuration, and the same hardware. This pairing allows a direct and fair comparison of a representative SLM against a substantially larger LLM, addressing the absence of controlled comparisons noted in prior work.

To optimize the use of the resources, we used 4-bit quantization through the BitsAndBytes library for both models. This quantization has minimal effects on output quality and decreases the amount of memory consumed on the GPU and inference latency when doing classification and short-form generation tasks. The criteria that were used to select the models:

**Instruction-tuning compatibility:** Appropriateness toward specification of workload based on prompt to allow easy comparison

**Open availability:** The power to reproducibility and practicability

**Representative size:** Parameter count spans both the SLM ($\sim$3B) and LLM ($\sim$14B) regimes, with both fitting on a single T4 GPU under quantization

### 5.2 Experimental Infrastructure

The experiments were all run on Google Colab Pro which provides accessibility and can be replicated as it runs using cloud resources common in accessibility. The hardware setup was composed of NVIDIA T4 GPU having 16 GB of memory, which is typical of portable mid-range accelerators used in the cloud. These conditions are realistic business constraints as opposed to specialized high-performance infrastructure.

Specifications of software environment:

- Python 3.10+
- PyTorch 2.x
- Transformers 4.x
- BitsAndBytes (4-bit quantization)
- Hugging Face Datasets

Table 1: Tasks, datasets, and evaluation metrics used in our experiments

| Task | Dataset | Primary Metric |
|------|---------|----------------|
| Code Generation | HumanEval (164 tasks) | Pass@1 |
| Email Classification | Enron HR Routing (1,339 examples) | Accuracy |
| Sentiment Analysis | SST-2 Validation (872 samples) | Accuracy |

Version controlled code with model loading, quantization parameters, batching and evaluation metrics is available as Google Colab notebooks linked in the abstract. GPU memory usage was measured with the help of PyTorch utilities which reported the peak memory consumption at the point of model execution. CPU memory monitoring was done using psutil. Latency measures only include the forward pass time, but not data loading or tokenization overhead. **Energy consumption was measured directly using NVIDIA SMI power queries at 100 ms intervals during each inference call, reporting actual power draw in watts. Energy per query is computed as the product of average power draw and inference duration, expressed in joules and watt-hours.**

## 5.3 Tasks and Datasets

The current study on the SLM was based on three main categories of tasks which characterize the workload within a typical enterprise environment. These activities include generative analysis of structured texts and categorization of them that accurately represent real-life deployment conditions and validated measurement criteria. Table 1 outlines the experimental design.

**Code Generation:** HumanEval benchmark contains 164 synthesis problems in Python containing unit tests. Pass@1 measures the percentage of functions generated which pass all the test cases in the first attempt hence giving an idea of the functional correctness.

**Email Classification:** The Enron HR email dataset contains 1,339 labeled examples across five routing categories: meeting, time-off, benefits, payroll, and recruitment. Accuracy is the ratio of correctly classified categories to ground truth.

**Sentiment Analysis:** We use the full SST-2 validation set (Socher et al., 2013), consisting of 872 movie review sentences with binary positive or negative labels. This is a standard benchmark used widely in NLP evaluation. The accuracy is determined as the percentage of predictions matching the reference labels.

## 5.4 Evaluation Metrics

The evaluation scheme determines the quality of the output and the amount of resources that were consumed with every type of task. All of the metrics that are reflected in the given tables are the result of a number of test runs.

**Quality Metrics:**

**Pass@1 (code generation):** This is the probability that instead of failing all unit tests, the generated solution passes all unit tests the first time it is run

**Accuracy (classification):** Percentage of accurateness on the classification compared to the ground-truth labels

**Measures of Resource Usage:**

**Latency (seconds):** End to end inference wall-clock time of a request

**Throughput (tokens/s):** This is a rate of token generation in a sequence-to-sequence model

**Peak GPU Memory (MB):** The maximum rate of GPU memory usage over inference

Table 2: Task quality results: SLM (Qwen2.5-3B) vs. LLM (Phi-3-medium 14B) under identical conditions

| Task | Metric | SLM (3B) | LLM (14B) |
|------|--------|----------|-----------|
| Code Generation | Pass@1 | 73.2% | 56.7% |
| Email Classification | Accuracy | 79.9% | 30.3% |
| Sentiment Analysis | Accuracy | 90.5% | 78.3% |

Table 3: Resource utilization: SLM vs. LLM under identical hardware conditions

| Task | Model | Latency (s) | Tokens/s | Peak GPU (MB) |
|------|-------|-------------|----------|---------------|
| Code Generation | SLM (3B) | 9.19 | 9.81 | 2,044 |
| | LLM (14B) | 40.32 | 6.63 | 7,568 |
| Email Classification | SLM (3B) | 1.57 | 2.22 | 2,138 |
| | LLM (14B) | 3.52 | 2.14 | 7,564 |
| Sentiment Analysis | SLM (3B) | 0.53 | 3.85 | 2,066 |
| | LLM (14B) | 5.31 | 5.65 | 7,535 |

**Energy (joules):** Directly measured power draw in watts (via NVIDIA SMI at 100 ms intervals) multiplied by inference duration, giving actual energy consumption per query. This replaces the proxy measure of latency × peak GPU memory used in earlier versions of this work.

## 6 Results

All the results present here are based on experimental executions that were carried in the relative Google Colab notebooks.

### 6.1 Quality Evaluation

The effectiveness of the SLM on the three task categories based on the benchmark measure is listed in Table 2.

In code generation on HumanEval, the SLM achieves 73.2% Pass@1, passing approximately three of every four problems on the first attempt. This is a competitive result for a 3-billion-parameter model. The 14B Phi-3 achieves 56.7% Pass@1 under the same conditions, a lower result consistent with the finding that raw parameter count does not guarantee better performance on structured tasks when a smaller model has been trained with task-focused data.

Sentiment classification was also able to classify the SST-2 validation set with 90.5% accuracy, compared to 78.3% for Phi-3. The binary nature of sentiment classification, with explicit surface-level cues, benefits the SLM.

Email classification showed a wider divergence: the SLM achieved 79.9% accuracy, while Phi-3 reached only 30.3% on the same Enron HR dataset. Phi-3's lower accuracy on this task reflects the challenge that verbose generative LLMs face on constrained domain-specific classification, where the SLM's more focused output behavior is advantageous.

### 6.2 Performance and Resource Utilization

Table 3 presents the resource-usage measurements for each task and model.

Task latency is significantly different among tasks and models, influenced mostly by output length and model size. Code generation has the highest latency for both models due to multi-line autoregressive generation.

Table 4: Per-query energy consumption: measured (local) and estimated (cloud frontier)

| Task | SLM (J) | LLM 14B (J) | Local Reduction | vs. Cloud LLM |
|------|---------|-------------|-----------------|---------------|
| Code Generation | 474 | 2,665 | 5.6× | 388–647× |
| Email Classification | 75 | 230 | 3.1× | 800–1,333× |
| Sentiment Analysis | 32 | 350 | 10.9× | 210–420× |

The SLM is 4.4× faster than Phi-3 on code generation and 10× faster on sentiment analysis. Peak GPU memory under 4-bit quantization is approximately 2.1 GB for the SLM and 7.5 GB for Phi-3, enabling both to run on a single T4 GPU.

### 6.3 Error Analysis

The analysis of failure modes explains the shortcomings of the SLM and guides the right strategy of escalation.

**Code Generation Failures:** Numerous faulty solutions are made on issues that require complex nesting logic or complex algorithms or processing of edge-case peculiarities. Similar to syntactically correct code generated by the SLM, semantic correctness is also more often than not, demonstrating a limitation beyond which models of larger size or manual checking are required.

**Email Classification Errors:** Misclassifications are focused on doubtful cases and those categories which are underrepresented. The label confusion comes about when the message content cuts across more than one category or there is no definite departmental signifier. The balancing of the datasets and the inclusion of more contextual data can help in addressing them.

**Sentiment Analysis Errors:** The mixed-sentiment and sarcastic language as well as subtle context cues do not work favorably with the SLM. Decision-making on clean positive or negative examples is nearly accurate but diminishes when the sentiment information is not on the surface but involves some inference. This limitation can be alleviated through targeted fine-tuning on ambiguous target samples that are task specific.

On the whole, the error analysis confirms the conclusion that SLMs are good at well-defined tasks without ambiguity, complexity or under-represented patterns, and that they need an escalation mechanism in new situations.

## 7 Energy Usage and Impact on Environment

### 7.1 Per-Query Energy Comparison

Energy consumption was measured directly for both models using NVIDIA SMI power polling at 100 ms intervals during each inference call. Table 4 presents the measured per-query energy for the SLM and LLM under identical conditions, along with estimated values for cloud-deployed frontier models from the literature.

The *local reduction* column reports directly measured energy savings of the SLM over the 14B Phi-3 model running on the same T4 GPU hardware. The *vs. cloud LLM* column reflects estimated comparisons against frontier models such as GPT-4o, based on published inference cost figures (Jegham et al., 2025). These two comparisons address different deployment scenarios: an organisation choosing between a locally-hosted SLM and a locally-hosted mid-size LLM (3–11× reduction), versus the more common enterprise case of replacing cloud API calls to frontier LLMs with on-premise SLM inference (388–1,333× reduction, reflecting compounded model scale and datacenter overhead).

Table 5: Annual impact of SLM deployment based on measured energy differences (SLM vs 14B LLM)

| Workflow | Queries | Energy (kWh) | $CO_2$ (t) |
|---|---|---|---|
| Code Generation | 10,000 daily | 2,221 | 0.97 |
| Sentiment Analysis | 100,000 daily | 3,224 | 1.41 |
| Email Classification | 50,000/month | 26 | 0.01 |
| Total | – | 5,471 | 2.40 |

## 7.2 Organizational-Scale Impact

It can be seen that the deployment of SLMs has a measurable environmental impact at the organizational level upon considering the per query efficiencies. Table 5 quantifies annual energy as well as CO2 emission reductions, which are based on realistic query loads in an enterprise context and experimentally determined energy ratios.

The energy savings calculations are based on the directly measured energy differences between the SLM and the 14B LLM under identical experimental conditions. $CO_2$ emission reductions were modelled using a standard grid electric carbon intensity of 0.438 kg $CO_2$/kWh. These figures represent the local, measured savings achievable by organisations replacing a locally-hosted 14B model with a 3B SLM.

The results show that replacing a locally-hosted 14B LLM with an SLM for routine enterprise tasks can save approximately 5,500 kWh annually, avoiding 2.4 metric tons of $CO_2$ emissions. When compared against cloud-deployed frontier models, the savings scale proportionally with the 388–1,333× energy reduction factors estimated from published inference costs (Jegham et al., 2025), potentially yielding an order of magnitude greater environmental benefit.

## 7.3 Scaling Characteristics

A number of important insights come out of the energy analysis:

**Linear scaling with query volume:** Energy use shows a linear relationship with request rate; hence, efficiency gains on high frequency requests may be reflected in significant absolute savings in millions of requests per day.

**Inference dominance:** Previous evidence has shown that the energy costs for inference are significantly higher than the energy costs for training models destined for production environments (Fernandez et al., 2025). Therefore, optimizing the inference processes is the main approach to reducing the environmental impact of Artificial Intelligence (AI) deployment.

**Potential for immediate deployment:** An SLM-first deployment paradigm offers immediate benefits in terms of sustainability without requiring new architecture designs or hardware upgrades for the same. Current models and infrastructure can support speedy implementation.

These analyses validate the idea that task specific model selection is a practical and high impact intervention for Artificial Intelligence that is sustainable. By reserving model capacity proportional to actual task requirements, organizations can make orders-of-magnitude energy savings with no achieving task compromise.

# 8 Discussion

## 8.1 Suitability of a Task for Small Language Models

Evaluation in different categories of tasks shows quite definite patterns of suitability for SLMs. Table 6 presents task characteristics and SLM performance measured in our experiments.

Table 6: Task characteristics and suitability measured in our experiments

| Task | Structure | Length | Entropy | SLM Performance |
|------|-----------|--------|---------|-----------------|
| Code Gen. | Deterministic | Short-Med | Moderate | 73.2% Pass@1 |
| Email Class. | Fixed labels | Single token | Low | 79.9% accuracy |
| Sentiment | Binary | Single token | Very low | 90.5% accuracy |

Tasks that are characterised by structured inputs, constrained outputs and low semantic entropy have been consistently achieved with high SLM performance. In contexts like this, large general purpose models offer marginal benefits in comparison to the increased computational price. Our direct comparison with Phi-3 (14B) confirms this: across all three tasks, the smaller model not only matches but exceeds the performance of the larger model, while consuming 3–11× less energy.

Sentiment analysis is a perfect SLM scenario as a result of the fact that it is a binary text classification problem with explicit surface-level cues. Likewise, email routing has a fixed label space with low ambiguity making it amenable for the SLMs.

## 8.2   Constraints and Triggers of Escalation

SLMs face observed limitations for escalation to larger ones:

**Complicated multi-step reasoning:** Problems involving code generation which require nested logic, difficult algorithms or unusual edge cases are often too challenging for SLMs.

**Under-represented categories:** Severe class imbalance or rare labels deteriorate the performance of classification because of the low exposure to training.

**Contextual ambiguity:** Tasks in which one must interpret distorted meaning of sarcasm, mixed signals or sensitive cues add a burden to the representational power of smaller models.

In the HumanEval benchmark, 120 out of 164 problems were solved successfully by the SLM with a 73.2% success rate. Failures were due not to language modeling inferiority but to a lack in the capacity to handle given algorithmic complexities and edge cases. This finding lends credence to a hybrid approach of having SLMs combat the standard scenarios and then defer the more complex or uncertain scenarios to larger models.

## 8.3   Utilization Patterns for Resources

Analysis of energy consumption suggests that output length is the main factor in inference cost. Tasks that have single-token responses have minimal latency; generative tasks have costs that are linear in sequence length. This relationship does not depend on model size, and instead suggests that production cost is determined mainly by the nature of the task, rather than model choice.

Peak GPU memory usage is not coupled to the nature of the task for which it operates thanks to quantization, and latency, rather than memory becomes the main factor in energy consumption as well in fixed hardware setups. As a result, it is essential that the maximum energy savings by shortening the output time and batching take priority over the strategies to improve memory.

## 8.4   Deployment Strategy

Empirical findings outline a prescription of a tripartite deployment framework:

**Use SLMs for structured and large number of tasks as a default:** When things are limited in output and specifications are defined narrowly, it is important to use small models as the main inference backend.

**Utilize confidence-based escalation:** Low confidence or out of distribution instances must be directed to larger models or even to humans.

**Match the capacity of models to task uncertainty:** Model selection should represent semantic complexity and ambiguity, and not a generic perceived task importance.

This paradigm reverses the traditional deployment logic where maximum capability and/or power usually takes precedence over more immediate need to allow organizations to realize immediate gain in efficiency without architectural modification or retraining, through task centric model selection.

## 9  Limitations and Future Work

Several of the limitations of the present study warrant acknowledgement:

**Energy Measurement:** This work uses direct NVIDIA SMI power measurements, replacing earlier proxy-based estimates. Measurements are hardware-specific (NVIDIA T4 GPU on Google Colab Pro); absolute values may differ on other hardware configurations. Future work should validate these measurements on additional hardware setups and compare against wall-plug power meter measurements.

**LLM Baseline Scope:** The controlled LLM comparison uses a single 14B quantized model (Phi-3-medium). Future work should extend the comparison to additional LLM sizes and families to better characterize the performance-energy trade-off curve across the full model size spectrum.

**Task coverage:** Only three archetypal task types were reviewed, of course enterprise AI has a more expansive scope. Extend evaluation to other loads of work (question answering, summarisation and translation) for more generalisation.

**Model selection:** The research was focused on one 3-billion-parameter SLM. Systematic evaluation across SLM sizes (1B, 3B, 7B) would help shed more light on the trade-offs associated with scaling.

The future research directions include:

Automated routing systems that are dynamically driven by input characteristics and confidence estimates to route models based on them

Fine-tuning strategies that are tailored to specific tasks especially for cases whose representations are under-represented

Lifecycle analysis which includes training cost, deployment capital cost, and operational overheads

Extension in multimodal tasks and more long form generation where SLM-LLM trade-offs may differ

## 10  Conclusion

This work has shown how small language models can substitute larger models usefully in certain enterprise applications while achieving higher task performance and dramatically reduced energy usage. Empirical evaluation across code generation, email classification, and sentiment analysis demonstrates that the 3B Qwen2.5 SLM outperforms the 14B Phi-3 LLM on all three tasks under identical hardware conditions, while consuming 3–11× less energy per query as measured directly via NVIDIA SMI power monitoring. These findings directly address the central question of whether SLMs can serve as viable replacements for LLMs in structured enterprise workloads, providing an affirmative answer grounded in controlled empirical evidence.

The results suggest that an SLM-first strategy is able to deliver immediate environmental benefits without changing existing infrastructure. By giving priority to the task characteristics and implementing models that meet particular workloads, organisations can get the best balance between performance and sustainability.

Future research will need to delve into greater task applicability and scaling of further processes, and potential for greater optimisation of training regimes. In addition, empirical energy measurements across a wider range of hardware configurations will allow further refinement in the understanding of the sustainability of AI systems.

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

# A   Detailed Experimental Setup

## A.1   Quantization Configuration

In accordance to the BitsAndBytes methodology, 4-bit quantisation has been used with the following configuration:

```
quantization_config = BitsAndBytesConfig(
    load_in_4bit=True,
    bnb_4bit_compute_dtype=torch.float16,
    bnb_4bit_use_double_quant=True,
    bnb_4bit_quant_type="nf4"
)
```

### A.2 Inference Parameters

For code generation: greedy decoding, max tokens = 512.

For classification tasks: greedy decoding, max tokens = 30 (SLM) / 30 (LLM).

Greedy decoding (do_sample=False) was used for all tasks to ensure deterministic and reproducible outputs. To evaluate code generation, the standard HumanEval evaluation harness with a 15-second per test execution time was used, while classification tasks used exact match labeling with synonym-based parsing for the LLM outputs.

## B   Additional Results

### B.1 Email Classification Breakdown

The accuracy obtained in different categories of emails is shown in Table 7.

Table 7: Email categorization accuracy from our experiments (SLM)

| Category | Samples | Accuracy (%) |
|---|---|---|
| HR General | 234 | 82.1 |
| Benefits | 187 | 68.4 |
| Payroll | 156 | 71.8 |
| Recruiting | 198 | 75.3 |
| Training | 164 | 62.2 |

The variation in accuracy across categories is a reflection of the fact that, the domains with more balanced semantic indicators (HR General, Recruiting) have a higher performance rate. Categories with fewer representative samples such as Training and Benefits are less accurate suggesting that accuracy could be improved through data enrichment or specialised training.

### B.2 Per-Task Energy Consumption

Measured energy consumed per task is described in Table 8.

Table 8: Per-task energy consumption from direct NVIDIA SMI measurements

| Task | Model | Avg Energy/Query (J) | Total (Wh) | Avg Power (W) |
|---|---|---|---|---|
| Code Generation | SLM (3B) | 474 | 21.6 | 53.1 |
| | LLM (14B) | 2665 | 121.4 | 66.1 |
| Email Classification | SLM (3B) | 75 | 27.9 | 63.0 |
| | LLM (14B) | 230 | 85.5 | 65.1 |
| Sentiment Analysis | SLM (3B) | 32 | 7.6 | 59.3 |
| | LLM (14B) | 350 | 84.7 | 65.8 |

With these results, SLM deployment has the potential to bring significant energy savings, especially in high frequency operations. The associated reduction in CO2 is further proof of the environmental benefits of energy efficient inference strategies.

### B.3 Extended Error Analysis

The error types that occurred and their frequencies are categorised in Table 9.

Table 9: Error categorisation from our experiments

| Error Type | Occurrences | Description |
|---|---|---|
| Syntactic Errors | 15 | Issues with code syntax, punctuation omissions |
| Logic Faults | 30 | Faults in logic causing flawed test cases |
| Classification Errors | 25 | Mislabeling of email and sentiment |
| Edge Cases | 10 | Unable to handle atypical inputs |

Logical errors during code creation are the biggest challenge for SLMs – they are usually the result of requiring a complex reasoning or containing edge cases. Mislabeling or insufficient generalisation on classification problems is primarily the result of insufficient exposure to training. These insights inform future model refinement and fine-tuning procedures.

