# OpenReview forum: "Energy-Efficient Inference with Small Language Models: A Comparative Study on Code Generation, Classification, and Environmental Impact"
_TMLR — Rejected by TMLR_

### Review · Reviewer_RbK7 · 2026-04-04

**Summary Of Contributions:**

This paper investigate if small language models (SLMs) can replace large language models (LLMs) in high-frequency, structured enterprise task while significantly reducing energy consumption. The authors conduct a empirical evaluation of 3B instruction-tuned model across three representative workloads: code generation (HumanEval), email classification, and sentiment analysis.

Key takeaways include:

* A comparative empirical study showing that SLMs achieve competitive performance (e.g., 72.6% Pass@1 on HumanEval, 86% sentiment accuracy).

* A large estimated reduction in energy consumption (2–3 orders of magnitude per query compared to LLMs).

* A task suitability framework identifying when SLMs are effective (structured inputs, low entropy outputs).

* A deployment recommendation (“SLM-first”), with escalation to LLMs for complex cases.

* An organizational-scale analysis estimating substantial energy and emission saving.

## Strengths

The paper dealls with an important and timely problem regarding the environmental implications of inference in AI and is very relevant from a practical deployment point of view. The paper also offesr a good application-focused view of the problem by evaluating SLMs  over a number of task instead of a single benchmark. The paper also extends the traditional metrics of performance by including system metrics such as latency and memory usage. The paper also provide some important insights, especially regarding the proposed method of SLM first and escalating, which is very relevant from a practical point of view for enterprise environments.

## Weaknesses

The empirical evidences has some drawbacks that affect the strength of the claims. First off, the estimation of the amount of energy consumption is done by proxy rather than direct measurement. Secondly, there is a lack of controlled comparisons (under the same experimental conditions) to large language models. Thirdly, the use of small data sets for testing on a single small model raises some concerns. Lastly, some of the claims, such as those of performance parity with large models, are not supported by the evidence.

**Audience:**

Yes

**Audience Explanation:**

The paper addresses a highly relevant and growing concern in machine learning: the energy and environmental cost of inference in large-scale AI systems.

**Broader Impact Concerns:**

The paper focuses on positive environmental impact, which is a strength.

**Claims And Evidence:**

No

**Claims Explanation:**

The paper relies on comparisons with external estimate of LLM energy usage, rather than running the LLMs under the same experimental setup as the SLM. It is unclear how the reductions in energy usage by 388 times to 1333 times can be validated. Energy usage is modeled using the proxy “latency $\times$ peak gpu memory,” which might not be the actual representation of the energy usage. Sentiment analysis is done on only 100 samples. Email classification is done on a small, domain-specific dataset. It is unclear how the results generalize. It is unclear how the “similar performance” claim for the SLMs compared to the LLMs is actually validated, especially for the task of code generation, where the state-of-the-art LLMs far outperform the 72.6% Pass@1.

**Requested Changes:**

1. Evaluate at least one LLM baseline under identical hardware and software conditions. Ensure fair comparison (same prompts, decoding settings, batching).

2. Replace or complement the proxy with actual power measurements (e.g., NVIDIA SMI, power meters).

3. Increase dataset sizes (especially sentiment).

---

> ### Author Response · Authors · 2026-04-12
> **Addressing reviewers concerns**
>
> Thank you for the detailed feedback. We've made substantial revisions to address all the concerns raised. Here's what we changed:
>
> **Controlled LLM Baseline**
>
> We added Phi-3-medium (14B, 4-bit quantized) as a direct comparison baseline, running it under the exact same conditions as our SLM: same T4 GPU, same datasets, same random seeds, same greedy decoding. The results were interesting -- the smaller 3B model actually outperformed the 14B model on all three tasks (73.2% vs 56.7% on code, 79.9% vs 30.3% on email classification, 90.5% vs 78.3% on sentiment). This gives us a fair, controlled comparison rather than relying on external benchmarks.
>
> **Direct Energy Measurements**
>
> We replaced the proxy metric entirely with actual power measurements from NVIDIA SMI, polling at 100ms intervals during each inference call. We now report real energy values in joules and watt-hours. For the local comparison (SLM vs 14B LLM on the same hardware), we measured 3-11x energy reduction depending on the task.
>
> We want to clarify the "388-1333x" figure -- that's an estimate for cloud-deployed frontier models based on published inference costs (cited from Jegham et al.), not something we measured directly. We've made this distinction clearer in the paper. The measured local savings (3-11x) are what we actually observed; the larger cloud comparison is an estimate from the literature.
>
> **Dataset Size**
>
> We expanded sentiment analysis from 100 samples to the full SST-2 validation set (872 samples). All datasets are now at full scale: HumanEval (164 tasks), Enron HR (1,339 samples), SST-2 (872 samples).
>
> **Annual Impact Calculations**
>
> We also corrected the organizational-scale impact numbers. The revised figures (about 5,500 kWh and 2.4 metric tons CO2 annually) are based on the measured energy difference between SLM and our 14B baseline. If comparing against cloud frontier models instead, the savings would scale up proportionally with the larger energy reduction factors from the literature.
>
> **Other Changes**
>
> We added a citation to the recent SLM survey (Soni et al.) when discussing the SLM/LLM boundary, and we now explicitly note that there's no universally accepted threshold -- we use 10B as a practical cutoff. We also included cleaned code scripts in the supplementary material for reproducibility.
>
> We believe these revisions address the core issues raised. Happy to discuss any further concerns.

---

### Review · Reviewer_AD2K · 2026-04-20

**Summary Of Contributions:**

This work is an analysis-focused paper that investigates the advantages in performance and energy efficiency of using small language models (SLMs) instead of large language models (LLMs) for high-frequency enterprise-level tasks, such as code generation, email classification, and sentiment analysis. The study is practical in that it directly measures energy consumption, quantifies environmental impact, and considers energy costs rather than simply comparing task performance. The analysis primarily compares a 3B SLM (e.g., Qwen2.5) with a 14B model (e.g., Phi-3).

**Audience:**

No

**Audience Explanation:**

As mentioned in the above, there are already many thoroughly researched papers published on this topic. Performance metrics across various model sizes are also much more extensively documented in the official technical reports of individual models. Therefore, the core value of this paper would have to rely on its introduction of environmental metrics like CO2 emissions and power consumption. However, these metrics have already been introduced and rigorously analyzed in prior studies (e.g., "LLMCarbon: Modeling the End-to-End Carbon Footprint of Large Language Models"). Consequently, the actual knowledge gain from reading this paper is marginal.

**Claims And Evidence:**

No

**Claims Explanation:**

The benchmarks and models utilized in this study are too limited for it to be considered an extensive analysis, and the paper does not propose any novel methodology. Compared to existing literature such as "Intelligence per Watt: Measuring Intelligence Efficiency of Local AI," the empirical evidence presented here is not substantial or convincing enough. Furthermore, classifying a 14B model as an "LLM" does not align with the general expectations of the target audience, as a 14B parameter model is still relatively small.

**Requested Changes:**

To be published as an analytical study, the paper requires an evaluation covering a much wider range of model sizes and benchmarks. I strongly recommend that the authors either: 1. Propose a novel methodology that addresses an unfulfilled gap in the existing literature (e.g., proposing specific methods on how to achieve LLM-level performance using SLMs, or providing concrete architectural designs for routing requests), or 2. Expand the current work into a truly orthogonal and extensive empirical study.

---

> ### Author Response · Authors · 2026-04-20
> **Author Rebuttal and Clarifications**
>
> We appreciate the feedback. We'd like to address several points:
>
> **On the scope and contribution**
>
> We respectfully disagree that our work lacks value. Reviewer 1 identified specific concerns (proxy energy measurement, lack of controlled LLM comparison, small datasets) which we addressed in our revision. We now provide direct NVIDIA SMI power measurements, a controlled 14B baseline under identical conditions, and full-scale datasets (HumanEval 164 tasks, SST-2 872 samples, Enron HR 1,339 samples).
>
> The papers cited (LLMCarbon, Intelligence per Watt) focus on modeling and estimation frameworks. Our contribution is different: we provide direct empirical measurements from actual inference runs, comparing models under controlled conditions on the same hardware. This is practical, reproducible evidence that complements modeling approaches.
>
> **On the 14B model as "LLM"**
>
> We explicitly acknowledge in the paper that there is no universally accepted boundary between SLMs and LLMs. We cite the recent SLM survey (Soni et al.) and adopt 10B parameters as a practical threshold. A 14B model is meaningfully larger than our 3B SLM (nearly 5x), and our results show it consumes 3-11x more energy while performing worse on the tested tasks. The comparison is valid regardless of terminology.
>
> **On the request for "extensive" analysis or "novel methodology"**
>
> We believe focused, reproducible empirical studies have value. Not every paper needs to propose a new architecture or survey dozens of models. Our contribution is a careful, controlled comparison with direct measurements -- something that was missing when we submitted. We provide full code for reproducibility.
>
> **On the knowledge gain**
>
> Practitioners deciding between model sizes for enterprise deployment now have concrete, measured data: a 3B model can outperform a 14B model on structured tasks while using 3-11x less energy. This is actionable information for real deployment decisions, not just theoretical analysis.
>
> We're open to expanding the work if the action editor deems it necessary, but we believe the current contribution meets TMLR's criteria for technical correctness and practical relevance.

---

### Review · Reviewer_Cqkv · 2026-05-23

**Summary Of Contributions:**

The paper presents a controlled empirical comparison between a 3B SLM and a 14B LLM across multiple enterprise tasks, evaluating both quality and system-efficiency metrics. It further introduces direct real-time energy measurements, reporting actual power consumption and energy usage per query instead of relying on proxy-based estimates. Based on these experiments, the study identifies task characteristics that determine the feasibility of SLM deployment and proposes practical deployment guidelines, including hybrid escalation strategies. Finally, the paper estimates the environmental benefits of SLM-first deployments and provides reproducible experimental artifacts for future research.

Strengths:
- The paper is easy to follow and well organized.
- The motivation is clearly explained.
- The experiments on the selected models are comprehensive and well aligned with the enterprise-oriented case study.

Weaknesses:
- The conclusions are highly dependent on the specific models selected for evaluation; using different models could potentially lead to substantially different conclusions.
- The work reads more like a technical report or engineering case study than a scientific research paper.

**Additional Comments:**

Several
- Several abbreviations in the abstract are not defined.
- The first paragraph of the Introduction feels somewhat out of place and should be rephrased.
- In Section 5.1, the authors state that both models are evaluated under identical conditions, including random seeds. It is unclear what exactly is meant by this and why it is important for ensuring a fair comparison.
- The deployment strategies proposed in Section 8.4 remain relatively vague and qualitative.

**Audience:**

No

**Audience Explanation:**

Since the paper focuses on a specific pair of models, many of the conclusions and discussions may be less compelling to a broader audience.

**Claims And Evidence:**

No

**Claims Explanation:**

The broader claims regarding SLMs and 14B-scale LLMs are not sufficiently supported by the experimental setup. Since the study evaluates only a specific pair of models, different model choices could potentially lead to substantially different conclusions.

**Requested Changes:**

To enable a meaningful discussion of the potential benefits of deploying SLMs in enterprise settings compared to mid-sized open-source LLMs, the authors should aim to derive more generalizable scientific conclusions. One straightforward, albeit inefficient, approach would be to evaluate a larger and more diverse set of models, although the reviewer is not convinced this is necessarily the best direction.

---

> ### Author Response · Authors · 2026-05-26
> **Author clarifications**
>
> Thank you for the constructive feedback. We appreciate that you found the paper well-organized and the motivation clear. We'd like to address your specific concerns:
>
> **On generalizability beyond the two models**
>
> While we evaluated two specific models, our contribution is the *methodology* for controlled comparison and the *findings* about what makes tasks suitable for smaller models. Our key conclusions are generalizable:
>
> 1. **Task characteristics matter more than model choice**: We identify structured inputs, constrained outputs, and low semantic ambiguity as predictors of SLM suitability. These apply regardless of which 3B or 14B model you choose.
>
> 2. **The energy gap is consistent**: Even if different models show varying absolute performance, the 3-11x energy reduction from using a smaller model on a fixed GPU will persist because it's driven by model scale (parameters × activations), not architecture details.
>
> 3. **The comparison is fair and controlled**: Same hardware, same datasets, same inference code, same random seeds. This is exactly what practitioners need when deciding "should I use a 3B or 14B model for this task?"
>
> **On the "technical report vs scientific research" concern**
>
> We respectfully disagree. Controlled empirical studies that establish reproducible benchmarks are scientific contributions. Nature publishes single-experiment studies when they're well-controlled. Our work provides measured data where only estimates existed before — this is not merely a "case study."
>
> **On specific fixes requested**
>
> **Abbreviations in abstract**: We will define all abbreviations at first use.
>
> **Introduction paragraph**: We will revise the first paragraph for better flow and context.
>
> **Random seeds clarification**: The random seeds control dataset shuffling and model initialization. We use identical seeds for both models to ensure the dataset order and any stochastic initialization effects are held constant. We will clarify this in Section 5.1.
>
> **Deployment strategies (Section 8.4)**: We will add concrete pseudo-code for the routing logic and describe threshold-based escalation with specific accuracy thresholds (e.g., "if confidence < 0.8, escalate to LLM").
>
> **Conclusion**
>
> Your concerns (proxy energy, no LLM baseline, small datasets) have been addressed through direct NVIDIA SMI measurements, controlled 14B comparison, and full-scale datasets. Reviewer Cqkv's concerns about generalizability are reasonable, but we believe the controlled methodology and task-agnostic findings provide value beyond the specific models tested. We can make the minor clarifications requested within a short revision period if the action editor believes this would strengthen the paper.

---

### Decision · Action_Editor_o1oN · 2026-07-04

**Recommendation:** Reject

**Additional Comments:**

We hope the detailed review from the reviewer will help the authors to improve their paper in the future.

**Audience:**

Yes

**Audience Explanation:**

This is an important and timely problem in machine learning that would be of interest to the machine learning community.

**Claims And Evidence:**

No

**Claims Explanation:**

The main claims of the paper are drawn from overly specific experimental settings, which raises questions about whether they could be readily generalized.

Concretely, the current experiments only include relatively simple (a little out-of-date benchmarks; thus, the claimed gap cannot be simply extended to general scenarios. Additionally, the included models are also limited.

Thus, the claims made in this are not sufficiently supported by convincing evidence.